# GraphAgent: Exploiting Large Language Models for Interpretable Learning on Text-attributed Graphs

## Abstract

This paper studies learning on text-attributed graphs, where each node is associated with a textual description. While graph neural networks (GNNs) have been widely employed for solving tasks on such graphs, they struggle with balancing between effectiveness and interpretability. Inspired by recent breakthroughs in large language models (LLMs), which have demonstrated remarkable capabilities with interpretable explanations across a variety of applications, we introduce GraphAgent. GraphAgent reframes learning on text-attributed graphs as an agent planning problem and parameterizes the agent as an LLM. This paradigm shift empowers the agent to take actions explicitly tailored for text-attributed graphs, enabling comprehensive exploration of both structural and textual features. Leveraging the expressive power of LLMs, the agent adeptly capture the intricate relationships inherent in the graph structure and textual descriptions, thereby yielding precise predictions and transparent reasoning processes. Extensive experiments conducted on various datasets underscore the effectiveness and interpretability of GraphAgent, shedding new light on the promising intersection of large language models and graph-based learning.

## 1 Introduction

Graph-structured data is a prevalent feature of the real world. A graph comprises nodes connected by edges, often accompanied by textual descriptions for each node, yielding a text-attributed graph (TAG). Modeling such type of data is a crucial topic within the field of machine learning, offering a variety of applications such as node classification and link prediction.

In the literature, graph neural networks have emerged as potent tools for modeling graph-structured data (Kipf & Welling, 2016; Velickovic et al., 2017; Gilmer et al., 2017). The core idea of graph neural networks is to acquire meaningful node representations through highly non-linear architectures, achieved via the message-passing mechanism. Specifically, each node's representation is iteratively updated by integrating information from both its own textual descriptions and those of its neighboring nodes. This fusion enables GNNs to harness the combined power of graph structures and textual features, consistently yielding state-of-the-art performance across numerous downstream applications. However, despite the effectiveness of GNNs in enhancing node representations, there is a downside. The message-passing mechanism can blend textual features from a multitude of nodes, resulting in highly unintuitive and uninterpretable representations. While recent efforts have sought to address this challenge (Huang et al., 2022; Zhang et al., 2021; Ying et al., 2019; Lin et al., 2020), these methods are still lack of an intuitive and transparent reasoning process.

In recent times, large language models (LLMs) have achieved remarkable success across various applications. Due to the use of vast text corpus for training, these models are endowed with a wealth of world knowledge and exhibit impressive emergent capabilities, as underscored by Wei et al. (2022). These inherent attributes empower LLMs to engage in intricate reasoning processes, yielding intuitive interpretations while tackling a wide array of tasks. Noteworthy examples include multi-hop question answering (Yao et al., 2022) and household planning (Singh et al., 2023). Furthermore, recent efforts have showcased LLMs as autonomous agents, adept at taking diverse actions and utilizing a variety of tools (Schick et al., 2023) to effectively address given problems, all the while

offering insights into their internal thoughts and thus demonstrating substantial interpretability. Inspired by the evidence, this paper endeavors to explore the application of LLMs in graph machine learning tasks, with the goal of achieving both effectiveness and interpretability.

For this goal, we introduce GraphAgent as our solution to leverage Large Language Models (LLMs) for graph machine learning tasks, with a specific focus on node classification due to its wide applications. One straightforward approach is to feed the graph structure and textual features of nodes into the memory of LLM for node label prediction. However, this method encounters challenges in real graphs, as they can be exceptionally large, and thus incorporating the entire contextual information into memory becomes infeasible or prohibitively resource-intensive. To overcome the challenge, we propose to reframe node classification as an agent-planning problem, with the agent parameterized by LLMs. Our LLM-powered agent is equipped with a few actions, including retrieving a node's features and accessing its neighboring nodes. This formalization empowers the agent to systematically explore the contextual information surrounding each target node and keep the most pertinent information within its memory. By striking a balance between exploration and exploitation (March, 1991), our approach optimizes memory utilization, ensuring efficiency on even extensive graphs.

Despite the agent-planning formalization, crafting an intelligent agent with desired effectiveness and interpretability remains challenging. GraphAgent surmounts this challenge by meticulously constructing illustrative examples for in-context learning. While a simplistic approach would involve manually crafting these examples, it comes at a high cost and lacks scalability. Consequently, we opt for a method that autonomously generates these demonstrations. Beginning from a blank memory, we let the agent generate trajectories on graphs at random, progressively incorporating trajectories culminating in accurate predictions as instructive examples into its memory for self-improvement. This iterative approach empowers the agent to evolve into a more effective node label classifier while simultaneously enhancing its ability to provide intuitive interpretations. Moreover, as the memory becomes increasingly crowded with the addition of more examples, we employ a hierarchical memory mechanism. This mechanism efficiently extracts and reuses the distilled insights gained from environmental feedback through a combination of long-term and short-term memory modules. In this manner, GraphAgent not only attains commendable effectiveness and interpretability but also manages to remain efficient in its operations.

We compare GraphAgent against supervised learning methods and in-context learning methods on three node classification datasets. Experimental results indicate that GraphAgent achieves comparable results to supervised learning methods. We also conduct ablation studies to validate our techniques and perform a comprehensive case study to analyze our successes.

## 2 RELATED WORK

### 2.1 GNN-BASED NODE CLASSFICATION

Our work is related to the field of graph neural networks, a widely adopted approach for modeling graphs. Prominent methodologies within this domain include Graph Convolutional Networks (GCN) (Kipf & Welling, 2016), Graph Attention Networks (GAT) (Velickovic et al., 2017), and Message Passing Neural Networks (MPNN) (Gilmer et al., 2017). At their core, these methods harness the message-passing mechanism to facilitate the learning of node representations. To achieve this, they iteratively refine the representation of each node by incorporating information from its own features as well as those of its neighboring nodes. In recent developments, there has been a notable trend towards integrating graph neural networks with pretrained language models (Zhao et al., 2022). This fusion has demonstrated enhanced capabilities in modeling textual node features within graph data. However, a persistent challenge in these approaches is the limited interpretability they offer, which can hinder a clear understanding of the model's decision-making processes.

In the quest to enhance the interpretability of Graph Neural Networks (GNNs), two predominant perspectives on eXplainable GNNs (XGNNs) have emerged. The first approach, known as blackbox interpretation, seeks independent methods to shed light on the relationship between GNN inputs and outputs. This perspective is exemplified by techniques such as GraphLIME (Huang et al., 2022) and RelEx (Zhang et al., 2021). The second approach strives to unravel the intricacies of GNNs by leveraging intrinsic information from GNN nodes and edges. Various methods within this category focus on subgraph structures to interpret GNN behavior. For instance, GNNExplainer (Ying et al.,

2019) identifies a compact subgraph and node features within a small subset, while PGMExplainer and GISST (Lin et al., 2020) generate task-relevant subgraphs and subsets of nodes to provide explanations. However, these methods often operate with a limited local scope and fall short of capturing the broader global context. In contrast, our proposed method adopts a more flexible approach to overcome this limitation through in-depth exploration of low-degree nodes. Additionally, (He et al., 2023) leverage Large Language Models (LLMs) as external knowledge sources to enhance node features, thereby facilitating downstream GNN finetuning. This combined approach enables us to provide comprehensive and contextually rich explanations within the GNN framework.

## 2.2 Large Language Models on Graph

LLMs have shown emergent abilities of reasoning as parameters scaling up, as observed in Wei et al. (2022). Transformer based LLMs like GPT-3 (Brown et al., 2020), PaLM (Chowdhery et al., 2022) make significant breakthrough. LLaMa (Touvron et al., 2023) outperforms GPT-3 and PaLM-540B while utilizing fewer parameters. Meanwhile, ChatGPT (OpenAI, 2022) achieves superior language understanding and reasoning capabilities. GPT-4 (OpenAI, 2023), the latest iteration, has showcased impressive performance across many tasks, including mathematics and coding (Bubeck et al., 2023). However, either training an LLM from scratch or finetuning one for graph learning (Ye et al., 2023) requires substantial computing sources. Consequently, we propose to formalize graph machine learning tasks as an agent-planning problem, and solve it with LLM-powered agents.

LLMs functioned as autonomous agents can interact with their environment using language instructions to solve various tasks (Weng, 2023). They possess zero-shot planning skills to decompose complex tasks into sub-tasks (Wang et al., 2023). LLMs can learn from trial and error (Shinn et al., 2023), but tasks like classification might not allow room for correction. These generative agents can maintain memory (Park et al., 2023), although they still face redundancy related to vector-based retrieval. ReAct equips LLMs with interactive capabilities (Yao et al., 2022) of reasoning and acting, but rigid demonstrations can limit capability of the agent to explore new strategies. AutoGPT excels in general problem-solving, but faces challenges in graph learning scenarios. Following these agent paradigm, we empower our agent with capabilities of planning, reasoning, acting and memory.

LLMs are facing great challenge on graph learning. Recent evaluations of LLMs focus on graph-to-text generation and graph classification (Guo et al., 2023; Yuan & Färber, 2023), revealing striking gaps with supervised SoTA methods. Enhancing node features with LLMs performs well on node classification tasks (He et al., 2023), but still relies on GNNs for prediction. Zhang (2023) finetunes LLMs to leverage external graph tools, including pretrained GNNs, to improve graph-related tasks. Another notable effort (Wei et al., 2023) employs an LLM agent to handle each stage in graph learning such as data preparation and architecture. However, training is still indispensable for these tool-based methods. To address these challenges, we propose a new prospective to view the graph as an environment where an agent can decide how to explore and use the available information.

## 3 Preliminaries

### 3.1 Node classification on Text-attributed Graph

Formally, a text-attributed graph $\mathcal{G}$ can be represented as $\mathcal{G} = (V, A, s_v)$ where $V$ is a set of nodes, where each node $v_i \in V$ represents an entity or concept. $A \in \mathbb{R}^{N \times N}$ is an adjacent matrix of nodes $V$ and $N$ is the number of nodes. $f_v$ denotes the text feature contains in node $v$. We study the node classification problem with **a few** labeled node $y_L$ which $L \subset V$. The goal is to predict the labels $y_U$ of unlabeled nodes $U = V \backslash L$. Specifically, $K$ is the number of labeled nodes of each category. Suppose $|Y|$ is the number of categories, we totally have $|L| = K \times |Y|$ labeled nodes.

### 3.2 Large Language Model for Node Classification

For our classification task with prompting, given the node $\mathcal{X}$, prompt $\mathcal{T}$ and LLM $p_{\text{LLM}}$, we denote the process of reasoning and decision making towards answers as $\mathcal{C}$. We aim to maximize the likelihood of label $\mathcal{Y}$ as:

$$p(\mathcal{Y}|\mathcal{T}, \mathcal{X}) = \sum_{\mathcal{C}} p_{\text{LLM}}(\mathcal{Y}|\mathcal{C}) p_{\text{LLM}}(\mathcal{C}|\mathcal{T}, \mathcal{X}) = \sum_{\mathcal{C}} \prod_{i=1}^{|\mathcal{Y}|} p_{\text{LLM}}(y_i|\mathcal{C}) \prod_{i=1}^{|\mathcal{C}|} p_{\text{LLM}}(c_i|\mathcal{T}, \mathcal{X}, c_{<i})$$

where $\mathcal{C}$ stands for steps of thought $s_t$ with action $a_t$ and graph observation $o_t$: $C = \{(s_t, a_t, o_t)\}_{t=1}^T$, $\mathcal{T}$ contains instructions and optional few-shot examples.

# 4 GRAPHAGENT: AUTOMATED TRAVERSAL CONTROLLED BY LLM

In this section, we introduce our approach GraphAgent, which is designed to enhance the effectiveness and interpretability of node classification within TAGs through the use of Large Language Models (LLMs).

To ensure robust interpretability, we reframe node classification as an agent planning task, employing LLMs to parameterize the agent. This agent is endowed with a repertoire of actions, such as retrieving neighboring nodes and accessing textual features of nodes. Following the chain-of-thought paradigm, we strategically design prompts that encourage the agent to articulate its reasoning process and execute the correct actions. The iterative reasoning steps, coupled with the actions taken at each step, yield a comprehensible interpretation of the final prediction made by the agent.

To augment the effectiveness of agents in node classification, we employ in-context learning and meticulously craft high-quality demonstration examples. This process starts with a less intelligent agent tasked with predicting labels for labeled nodes, resulting in a collection of trajectories, each comprising a sequence of thought-action-observation triplets. Then, we enrich the agent's memory with trajectories that end in correct answers, facilitating ongoing in-context learning. Through this approach, the agent consistently refines its capabilities, yielding more accurate predictions.

Now, let us delve into the specifics of the GraphAgent framework.

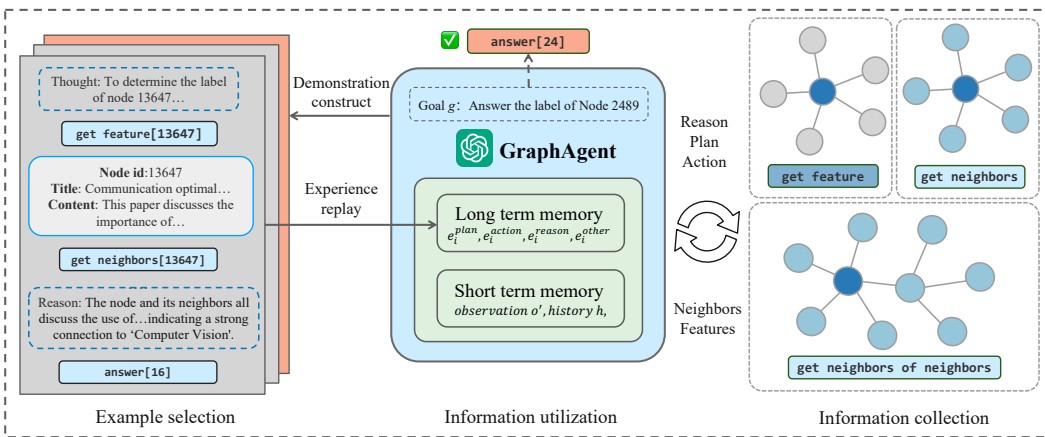

Figure 1: The GraphAgent framework is used for constructing demonstrations and predicting node labels. It employs a specialized agent, powered by LLM, to gather information from the graph through predefined actions. Decisions of predictions and actions are relying on the utilization of information residing in hierarchical memory structures. Check more details in Algo. 1
.

## 4.1 AGENT-PLANNING FORMALIZATION

A key innovation introduced by GraphAgent lies in its formalization of node classification as an agent-planning task. In formal terms, starting at the target node whose label needs to be predicted, the agent is equipped with a collection of actions to systematically explore the local context surrounding the target node. To enable the agent to effectively traverse both the structural and textual dimensions of nodes, we define the following actions:

**get_neighbors[nodeid]**: This action permits the agent to retrieve the first 10 related neighbors of the node [nodeid], facilitating an exploration of the structural information in the vicinity of the node [nodeid].

**get_feature[nodeid]**: This action provides access to the detailed textual features of the node [nodeid], enabling the agent to grasp the semantics associated with the node [nodeid].

Besides the above two actions, we also add the following special action:

**answer[label]**: This action enables the agent to output the predicted answer `[label]`.

Upon commencing the inference process, we initialize the memory of the agent with a system prompt. This prompt not only informs the agent about the permissible actions but also imposes certain constraints. Following the chain-of-thought method, we incorporate instructions that encourage the agent to articulate its step-by-step thought process during reasoning. Subsequently, the agent performs a series of decision-making steps. At each step, denoted as $t$, the agent deliberates based on its current memory $\mathcal{M}_t$ and produces both a string of thought $s_t$ and an action $a_t$. This action $a_t$ is then executed, yielding an observation $o_t$. The step concludes with the update of the agent's memory, encompassing the thought $s_t$, action $a_t$, and observation $o_t$. This iterative process continues until the agent opts for the special **answer** action, signifying its readiness to deliver its final prediction.

## 4.2 IN-CONTEXT LEARNING

The previous section outlines our approach to formalizing node classification as an agent-planning problem. To empower the agent for effective node classification, we construct a set of high-quality examples to serve as demonstrations for in-context learning.

In this endeavor, we note that a set of labeled nodes is often available in node classification, and these labeled nodes can be used for example construction. Specifically, we begin with a simple agent solely fed with the system prompt without any examples for demonstration, and apply this agent to nodes within the labeled set. During each run at a node, the agent generates a trajectory, i.e., a sequence of $(s_t, a_t, o_t)$ triplets, where $s_t$ represents a string of thought, $a_t$ denotes an action, and $o_t$ stands for an observation. Each trajectory culminates in a prediction of the node's label. Intuitively, trajectories ending with the correct node labels are more instructive. Therefore, we selectively retain such trajectories, and for each trajectory we concatenate all the $(s_t, a_t, o_t)$ triplets to create an in-context learning example. This approach yields a substantial number of high-quality in-context learning instances.

However, the memory capacity of large language models is typically constrained, and it is impractical to accommodate all the collected examples for in-context learning. Therefore, we employ a neat strategy to identify and select the most crucial examples for this purpose. Specifically, in line with Liu et al. (2021), we retrieve semantically similar examples as demonstrations for each test node. Initially, we utilize a sentence encoder, denoted as $E$, to transform the node titles within the examples into sentence representations, denoted as $r_1, r_2, ..., r_n$. Subsequently, for each test node $x$, we compute the similarity between its sentence representation $r_x$ and the representation of each example. Besides, examples with more steps can potentially enhance reasoning as it presents a more thorough inference process, as demonstrated by (Fu et al., 2022), and thus the number of reasoning steps can serve as a heuristic reward for these examples. Putting all the intuitions together, we define a Similarity & Complexity score (SC) as follows:

$$SC = \text{sim}(E(\mathcal{G}_{\text{title}}(N_x)), E(\mathcal{G}_{\text{title}}(N_i))) + \frac{T_i}{\alpha} \tag{1}$$

In this equation, we factor in both the similarity between the title of the example node $N_i$ and the node to be classified $N_x$. Additionally, we include $\frac{T_i}{\alpha}$ as a reward term to encourage the inclusion of longer examples with more steps $T_i$. For each test node, the top-k examples with the highest SC scores are then utilized for subsequent in-context learning.

By employing this method for constructing and selecting in-context learning examples, our approach achieves a commendable balance between interpretability and effectiveness.

## 4.3 HIERARCHICAL MEMORY UTILIZATION MECHANISM

To further effectively convey a wealth of information within the constraints of limited context length, we implement a hierarchical memory utilization mechanism for in-context learning. This mechanism capitalizes on the fusion between two memory modules, i.e., the short-term memory module, denoted as $M_s$, and the long-term memory module, referred to as $M_l$, facilitating the processing of both long text from graph environment feedback and past interaction experiences.

**Short-term memory module $M_s$: Goal-oriented summarization**

We first indicate $g_i$ as the goal of classifying node $n_i$ at the start of each classification iteration. For each time step $t$, the current obtained information $o_t$ from graph environment's feedback was summarized by LLM: $p_{\text{LLM}}(o'_t|o_t, g_i)$ towards goal $g_i$ into $o'_t$. We append $o'_t$ as short term memory $\text{memory}_{\text{short}}$ instead to context after every action execution.

**Long-term memory module $M_l$: Experience replay**

We use a simple method to compress the multiple steps of each example $e_i$ as demonstrations for in-context learning: an LLM conditioned on classification goal $g_i$ summarize the example's trajectory $e$ from different perspective: plan designing $e^{\text{plan}}$, choices of actions $e^{\text{action}}$, reasoning process $e^{\text{reason}}$ and other points $e^{\text{other}}$ considered valuable by LLM into $p_{\text{LLM}}(e_i^{\text{plan}}, e_i^{\text{action}}, e_i^{\text{reason}}, e_i^{\text{other}}|e_i, g_i)$. We attach those experiences $\text{memory}_{\text{long}}$ extracted from examples before each classify iteration.

## 5 EXPERIMENT

### 5.1 EXPERIMENT SETUPS

**Datasets.** We perform our experiments on three pretigious benchmarks, spanning a range of scales from small to medium to large: Cora, ogbn-arxiv, and ogbn-products (Hu et al., 2020). Tab. 1 presents statistics of these datasets. We provide more details in App. C.

Table 1: Statistics of the datasets.

|  | #Nodes | #Edges | Avg. Node Degree | Class. Num | Split(%) |
|---|---|---|---|---|---|
| cora (Papers) | 2,708 | 5,439 | 2.0 | 7-class | 60 / 20 / 20 |
| ogbn-arxiv (Arxiv) | 169,343 | 1,166,243 | 13.7 | 40-class | 54 / 18 / 28 |
| ogbn-products (Products) | 2,449,029 | 61,859,140 | 50.5 | 48-class | 8 / 2 / 90 |

**Baselines.** We compare the performance with two prominent paradigms in graph machine learning, namely multi-Layer Perceptrons (MLP) and Graph Neural Networks (GNNs). For the MLP architecture, we consider CoLinkDistMLP (Luo et al., 2021), which leverages the distillation of knowledge from adjacent nodes into MLPs. On the GNN side of the spectrum, we incorporate the SSP model (Izadi et al., 2020), which has achieved state-of-the-art test accuracy on the Cora dataset through the optimization of GNNs using natural gradient descent. Additionally, we observe the remarkable performance of TAPE+RevGAT (He et al., 2023) on the ogbn-arxiv dataset, where it enhances the text features of nodes using LLMs. For the ogbn-products dataset, GLEM (Zhao et al., 2022) emerges as the top-performing model. For fair comparison, we establish in-context learning baselines for both GPT-3.5 (OpenAI, 2022) and GPT-4 (OpenAI, 2023), providing them with contents, neighboring nodes, and related labels. For further insight, we apply self-consistency (Wang et al., 2022) to GraphAgent (*gpt-3.5*). This approach involves generating multiple reasoning paths from LLMs and subsequently conducting a majority vote to determine the final answer. Moreover, we introduce a zero-shot GraphAgent (*gpt-4*), devoid of any prior demonstrations.

**Implementation details** We use two OpenAI language models: GPT-3.5[1] and GPT-4[2] through their APIs. Due to the significant cost associated with calling the OpenAI API, we employed a random sampling approach. We repeat selecting 50 data points separately from test and validation set for three times, each time with different random seeds. We report the mean and standard deviation of results for each configuration. For constructing demonstration, we generate $u_{exp} = 10$ examples for each category. We established the parameters "temperature" and "top_p" at 0.4 and 0.6, respectively. We set the complexity reward coefficient $\alpha = 20.0$ for optimal results and max generation steps $T_{max} = 5$ for efficiency. Lastly, we employed the Sentence Transformer[3] as an encoder for calculating sentence embeddings and assessing semantic similarity to retrieve few-shot examples.

### 5.2 MAIN RESULTS

Tab. 2 presents the main results of our study, highlighting distinct advantages of GraphAgent in the following key aspects:

---

[1] https://platform.openai.com/docs/models/gpt-3-5

[2] https://platform.openai.com/docs/models/gpt-4

[3] https://huggingface.co/sentence-transformers/all-MiniLM-L6-v2

**(i). GraphAgent is comparable with supervised methods.** Notably, even without a training process, our best results are approaching the SoTA within GNNs, with only a slight margin in test accuracy of 3.2% in Cora, 9.7% in ogbn-arxiv, and a 6.5% margin in ogbn-products. Also, GraphAgent outperforms MLPs on ogbn-arxiv and ogbn-products.

**(ii). GraphAgent enhances LLMs.** GraphAgent (*gpt-3.5*) outperforms in-context learning (ICL) methods powered by *gpt-3.5* with test accuracies of 7.44, 18.66 and 12.67 (↑15.0∼43.7%), compared to 10.00, 22.00 and 10.00 (↑18.1∼56.9%) in the validation phase. Moreover, even without any demonstrations, GraphAgent (*gpt-4*) surpasses ICL (*gpt-4*) by 12.66, 7.34 and 4.66 (↑6.1∼21.6%) in the test phase, compared to 11.34, 11.34 and 6.00 (↑7.6∼20.5%) in the validation phase.

**(iii). GraphAgent benefits from self-consistency.** Utilizing 10 sampling paths yields a substantial increase in test accuracy compared to a single path with improvements of 7.8%, 6.5%, and 15.0% in 3 datasets, respectively. In some cases, GraphAgent (*gpt-3.5*) even surpasses ICL (*gpt-4*) with 4.1% and 13.9% improvements on Cora and ogbn-arxiv. This enhancement through self-consistency suggests that further increasing the number of sampling paths in GraphAgent (*gpt-4*) could potentially mitigate the remaining performance gap to supervised state of the art.

Table 2: Node classification accuracy on three datasets (mean±std%) ICL: In-context learning. $K$ indicates number of examples while $P$ suggests sampling paths for self-consistency.

| Type | Method | Cora | | Arxiv | | Products | |
|---|---|---|---|---|---|---|---|
| | | test | val | test | val | test | val |
| MLP | CoLinkDistMLP | 87.54±0.00 | - | 56.38±0.16 | 58.07±0.07 | 62.59±0.10 | 77.21±0.15 |
| GNN | SSP | **90.16**±0.59 | - | - | - | - | - |
| | GLEM | - | - | 76.97±0.19 | 77.49±0.17 | **90.14**±0.12 | 93.70±0.04 |
| | TAPE+RevGAT | 89.90±1.11 | - | **77.50**±0.12 | 77.85±0.16 | - | - |
| LLM | *gpt-3.5-turbo* | | | | | | |
| | ICL | 49.33±1.16 | 51.33±4.16 | 42.67±4.16 | 38.67±1.16 | 54.00±2.00 | 55.33±4.16 |
| | GraphAgent$_{P=1}$ | 56.67±3.06 | 61.33±4.16 | 61.33±1.53 | 60.67±3.06 | 66.67±3.06 | 65.33±1.16 |
| | GraphAgent$_{P=10}$ | **61.13**±1.16 | **63.33**±1.16 | **65.33**±1.16 | **64.67**±1.16 | **76.67**±1.16 | **74.67**±3.06 |
| | *gpt-4* | | | | | | |
| | ICL | 58.67±1.16 | 59.33±1.16 | 57.33±3.06 | 55.33±2.31 | 76.67±4.16 | 78.67±3.06 |
| | GraphAgent$_{K=0}$ | 71.33±3.06 | 70.67±1.16 | 64.67±1.16 | 66.67±2.31 | 81.33±3.06 | **84.67**±1.16 |
| | GraphAgent$_{K=3}$ | **87.33**±2.31 | **87.33**±1.16 | 70.67±2.31 | 69.33±1.15 | **84.67**±2.31 | 83.33±1.16 |

## 5.3 CASE STUDY: REASONING PROCESS

**Keyword extraction.** The GraphAgent typically begins by initiating the `get_feature` action, aimed at obtaining the content of a specified node. Subsequently, it conducts an analysis of the feedback content through keyword extraction. As illustrated in Fig. 2a, when examining the content retrieved from `node 51697`, GraphAgent identifies keywords such as *political alignment* and *political messaging*, which are highly indicative of the *Computers and society* category. Thanks to the extensive commonsense knowledge embedded within the LLM, this often leads to the discovery of the answer or, as an alternative, aids in narrowing down the potential answers within a candidate set.

**Potential label analysis.** Neighbors provide vital information in graph (Angelova & Weikum, 2006). Continuing its quest for answers, GraphAgent tends to explore the neighbors of the node to gather additional information through the `get_neighbors` action. Fig. 2b shows an example where GraphAgent judges potential category of each obtained neighbor's content. Then, GraphAgent refines the potential label options based on common features shared among these neighbors.

**Deep exploration for low-degree node.** For lower-degree (3-degree) `node 160080`, as depicted in Fig. 2c, the information available from its neighbors is limited. To tackle this challenge, GraphAgent devises a strategy to explore the neighbors of its neighbors. This extended exploration enables GraphAgent to infer the label of `node 127272` by capitalizing on their shared *distributed learning* feature. This strategy empowers the agent to expand its analysis, encompassing larger sub-graphs within sparse regions for the determination of specific characteristics within this portion.

**(a).Key word extraction**

**Action:** get_features[51679]
**Title:** Ad delivery algorithms the hidden arbiters of political messaging
**Content:** Political campaigns are using digital advertising to target voters, but ad delivery algorithms can lead to demographic skews without the advertiser's knowledge. Facebook's ad delivery algorithms differentiate the price of reaching a user based on their inferred political alignment, hindering campaigns' ability to reach voters with diverse political views. This can contribute to political polarization and filter bubbles. Reforms to restrict targeting tools will not be enough to ensure ads reach users with diverse views. There is a need for more transparency in the political advertising ecosystem.
**Reasoning:** The content mentions 'ad delivery algorithms', 'political messaging', 'political campaigns', 'political alignment', and 'political advertising ecosystem', which are related to the field of 'Computers and Society'

**(b).Potential label analysis**

**Action:** get_neighbors[149489]
**Neighbors:** 28309: tom net learning transparent object matting from a single image
67166: deep residual learning for image recognition
34064: u net convolutional networks for biomedical image segmentation
104194: deep multi scale convolutional neural network for dynamic scene deblurring
**Action:** get_feature[28309,67166,34064,104194]
**Content:** ...104194 The paper discusses Conventional methods for deblurring rely on simple assumptions and...
Reasoning: The neighbor with node_id 104194 is related to deep multi-scale convolutional neural networks for dynamic scene deblurring This suggests that Node 149489 might also be related to computer vision or image processing.
**Answer:** Based on the topics discussed by the neighbors (transparent object matting, image recognition, biomedical image segmentation, and dynamic scene deblurring), it is likely that Node 149489 is related to 'Computer Vision'.

**(c).Low degree exploration**

**Action:** get_features[160080]
**Title:** adaptive stopping rule for kernel based gradient descent algorithms
**Action:** get_neighbors[160080]
**Neighbors:**
127272: distributed learning with regularized least squares
32137: lazysvd even faster svd decomposition yet without agonizing pain
35253: revisiting the nystrom method for improved large scale machine learning
Thought: I can't reason our the common feature. Further analyze the labels and features of the neighbors can provide insights into the potential label of Node 160080.
**Action:** get_features[127272]
Title: distributed learning with regularized least squares
Plan: I have obtained the details of one of the neighbors of Node 160080 using the 'get_features' command. By analyzing the neighbors of this neighbor, I can gather more information to help determine the label of Node 160080.
**Action:** get_neighbors[127272]
**Neighbors:**
82873: optimal distributed online prediction using mini batches
40283: convergence of online mirror descent algorithms
85108: fast and strong convergence of online learning algorithms
55142: max diversity distributed learning theory and algorithms
67631: distributed learning with dependent samples
167509: distributed learning with random features
**Reasoning:** neighbors of Node 127272 talk about distributed learning and convergence of learning in common, which are topics related to Machine learning.
**Answer:** The content of Node 160080, 'adaptive stopping rule for kernel based gradient descent algorithms', indicates that it is related to machine learning. Additionally, its neighbors also have machine learning-related titles Therefore, I believe the label of Node 160080 is 24: Machine Learning.

Figure 2: Case study of GraphAgent on ogbn-arxiv: Three common strategies employed by LLMs to reach the final answer with an explicit reasoning path. Key factors influencing the results are highlighted in yellow text, with the correct answer indicated in green highlight.

## 5.4 ANALYSIS OF IN-CONTEXT LEARNING EXAMPLES

We delve deeper into the impact of varying the number of sampling paths $P$ and examples $K$ with GraphAgent (*gpt-3.5*) in zero-shot manner. As shown in Fig. 3, our findings reveal a clear trend: increasing both $P$ and $K$ have positive effect on accuracy. Remarkably, the involvement of just a single example results in accuracy improvements of 0.28, 0.26, 0.31 and 0.32, for different values of $P$ (specifically, $P = 1, 5, 10, 20$). This highlights the ongoing challenge posed by graph-based learning tasks, particularly in scenarios where no prior demonstrations are available. Notably, as we increment $P$, we observe a gradual improvement in accuracy. We suggest increasing the number of sampling paths can reduce occasional deviations in LLMs' API responses, thus leading to improved accuracy. However, this improvement tends to reach a plateau when $P$ is continually increased.

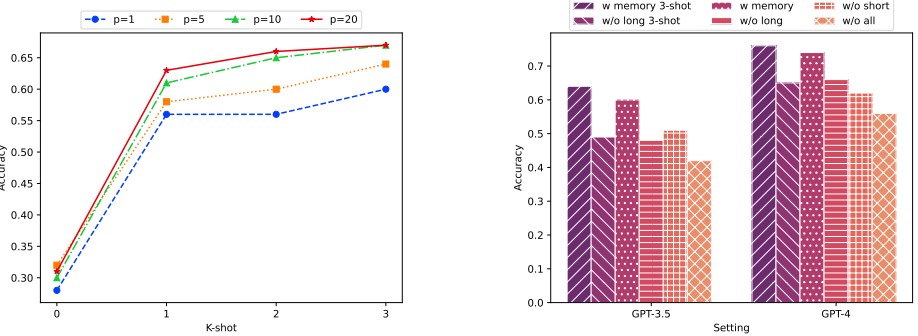

Figure 3: Analysis of self-consistency sampling path $P$ and example quantity $K$. Note that $P = 1$ equals to without self-consistency.

Figure 4: Ablation study of memory mechanism on ogbn-arxiv. Unless otherwise stated, all experiments are performed in a 1-shot setting.

## 5.5 ABLATION STUDIES OF DIFFERENT COMPONENTS

**Short and long-term memory work together.** As shown in Fig. 4, compared with non-memory, the hierarchical memory mechanism boosts accuracy from 0.42 to 0.60 (↑42.9%) with GPT-3.5 while

from 0.56 to 0.74 (↑32.1%) with GPT-4. Solely using long-term or short-term memory makes the performance decrease 15.0% and 20.0% with GPT-3.5, 16.2% and 10.8% with GPT-4 respectively. We found that long-term memory improves more performance as number of examples increasing, enhancing 26.9%(3-shot) > 24.0%(1-shot) with GPT-3.5 and 16.7%(3-shot) > 12.1%(1-shot) with GPT-4. Note that we use stronger *gpt-3.5-turbo-16k* to hold longer context when 3-shot and without long-term memory. However, it still fall 18.3% behind 1-shot with full memory mechanism.

Figure 5: Successful experience and failure experience replayed by long-term module $M_l$. GraphAgent analyzes successful reason and reflects upon mistakes.

**Successful examples experience brings boost.** To determine the reflective ability of GraphAgent, we adjust our experience replay strategy to learn from fail examples: as shown in Fig. 4, we enrich the example store $\mathbb{E}$ with generated failed examples to let GraphAgent reflect why they made mistake at first trail. As depicted in Tab.3, it is evident that when we include failure cases, the accuracy significantly decreases in both the ogbn-arxiv and ogbn-product, in comparison to solely relying on successful experiences. Notably, with the exception of random selection in ogbn-products, the inclusion of failure cases results in a performance improvement of 3.6%. We posit that this improvement can be attributed to the fact that random sampling is not an inherently optimal selection strategy.

Table 3: Analysis of Example Selection Strategies: *-diff %* indicates decrease of accuracy, while $\Delta\%$ represents the drop in performance compared to **SC**(*ours*). *Success* indicates only selection from successful experiences, whereas *w/ failure* enriches the example store with failure cases.

|  |  | **Rigid** | $-\Delta$ % | **Random** | $-\Delta$ % | **-Complex** | $-\Delta$ % | **SC** |
|---|---|---|---|---|---|---|---|---|
| | *Success* | 42.0 | 32.3 | 47.0 | 24.2 | 55.0 | 11.3 | 62.0 |
| Arxiv | *w/ failure* | 35.0 | 16.7 | 42.0 | 22.2 | 51.0 | 5.6 | 54.0 |
| | *-diff %* | 16.7 | - | 10.6 | - | 7.2 | - | 17.7 |
| | *success* | 50.0 | 30.6 | 56.0 | 22.2 | 64.0 | 11.1 | 72.0 |
| Product | *w/ failure* | 44.0 | 34.3 | 58.0 | 13.4 | 61.0 | 8.9 | 67.0 |
| | *-diff %* | 12.0 | - | -3.6 | - | 4.9 | - | 6.9 |

**Example sampling strategy.** We compare three different example sampling strategies to our Similarity & Complexity-based (**SC**) search method. The **Rigid** approach entails fixing a random example during each test iteration. In the **Random** approach, we randomly select an example from the store denoted as $\mathbb{E}$ for each test item. In the **-Complex** setting, we remove the reward term $\frac{T_i}{\alpha}$ from F.1 to see the difference without encouragement for complex example. Our findings consistently demonstrate that our **SC** selection strategy outperforms the other three settings, taking into account both similarity and complexity.

## 6 CONCLUSIONS

In this paper, we delve into the potential of leveraging LLMs as agents in graph machine learning. We propose GraphAgent, which reframes graph learning as an agent-planning problem and parameterizes the agent with LLMs. A neat strategy is proposed to construct demonstration examples for in-context learning, allowing GraphAgent to well balance effectiveness and interpretability. We conduct extensive experiments to evaluate GraphAgent. Results show that GraphAgent achieves comparable results to SoTA supervised methods while demonstrates intelligent and transparent reasoning process. Our experiments also emphasize the potential to close the performance margins by incorporating more sampling paths. We anticipate that our work will spark innovative investigations, driving the evolution of interpretable and intelligent agents in graph machine learning.

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
