# A  PSEUDOCODE OF GRAPHAGENT

---

**Algorithm 1** GraphAgent inference algorithm

---

**Require:** initial instruction $p_{\text{init}}$, long term prompt $p_{\text{long}}$, shor term prompt $p_{\text{short}}$
**Require:** Language model LLM, Sentence encoder $E$
**Require:** Graph $\mathcal{G}$, node to be classified $N_c$, goal description $g$
  **for** $e_i$ in example store $\mathbb{E}$ **do**
    score $c_i \leftarrow \text{sim}(E(\mathcal{G}_{\text{title}}(e_i)), E(\mathcal{G}_{\text{title}}(N_c))) + \frac{T_i}{\alpha}$
  **end for**
  **find** $k$ examples $list_e = [e_1, .., e_k]$ with top_k score
  **for** $e_j$ in top_k examples $list_e$ **do**
    $memory^j_{\text{long}} \leftarrow M_l(\text{LLM}, p_{\text{long}}, e_j, g)$
  **end for**
  **while** $t < T_{max}$ **do**
    response $\mathbf{r_t} \leftarrow \text{GraphAgent}(\text{LLM}, p_{\text{init}}, memory^1_{\text{long}}, \cdots, memory^k_{\text{long}}, memory_{\text{short}})$
    **if** `extract_action`$(\mathbf{r_t})$ == **answer then**
      **return** category id `extract_label`$(\mathbf{r_t})$
    **end if**
    observation $o \leftarrow \text{execute}(a_i, \mathcal{G})$
    $memory_{\text{short}} \leftarrow M_s(\text{LLM}, p_{\text{long}}, o, g)$
    $t \leftarrow t + 1$
  **end while**
  **return** category id `extract_label`$(\mathbf{r_t})$

---

# B  MORE ABLATION STUDIES

## B.1  ABLATION STUDY OF ACTION DESIGNING

To verify the validity and necessity of our action space design, we conduct ablation experiments on our action selection. Note that the **answer** must be remained for giving the prediction. We then remove **get_feature[node_id]** and **get_neighbors[node_id]** action separately to see accuracy changes. Also, we add another relevant action **analyze_label[node_id]** to test the result. Check designing details in Sec. D.5.

Tab. 4 evidently show that removing both **get_feature** and **get_neighbors** declined the results severely. In *w/o all* settings, we remove all actions except **answer**. Note that, the execution typically fails in coming into termination with generation of illegal actions such as 'neighbors' or 'feature'. Which also indicates requiring neighbors and feature is necessary for node classification. In other cases, LLMs can only randomly guess from title of the node. When adding with **analyze_label** action, LLM tends to use this action over and over again and ignoring information collection of the graph. LLM think **analyze_label** as the closest to its intention, however, this action is too general to classify the node.

Table 4: Ablation study on action design, we remove all predefined actions in *w/o all* setting

|  | ogbn-arxiv | ogbn-product |
|---|---|---|
| *full* GraphAgent | 29.67 | 53.33 |
| *w/o* **get_feature** | 26.33(↓11.3%) | 45.33(↓15.0%) |
| *w/o* **get_neighbors** | 22.33(↓24.7%) | 35.67(↓33.1%) |
| *w/o all* | 6.33(↓78.7%) | 12.67(↓76.2%) |
| *w* **analyze_label** | 18.67(↓37.1%) | 25.33(↓52.5%) |

Table 5: Ablation study on reasoning and planning process of GraphAgent.

| reason | plan | ogbn-arxiv |
|:---:|:---:|:---:|
| ✗ | ✗ | 22.33(↓24.7%) |
| ✓ | ✗ | 25.67(↓13.5%) |
| ✗ | ✓ | 25.33(↓11.8%) |
| ✓ | ✓ | 29.67(-) |

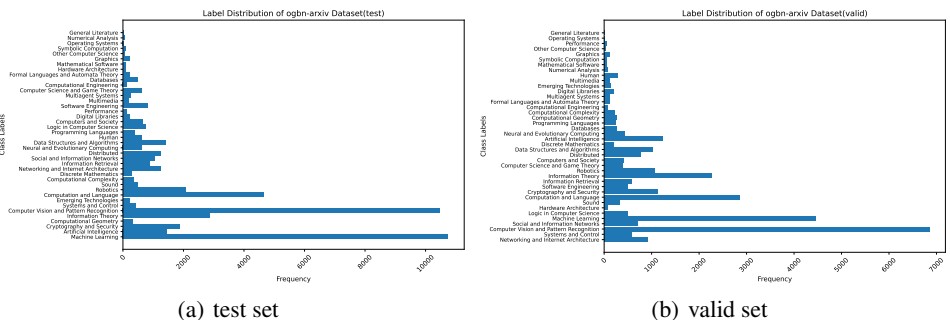

(a) test set          (b) valid set

Figure 6: Label distribution in ogbn-arxiv

## B.2 ABLATION STUDY OF REASONING PROCESS

We remove the explicit process of reasoning and planning separately on arxiv dataset in zero-shot manner, we simply sample 150 items from train set. Here as shown in Tab. 5, we found that the thought of reasoning and planning both enhances the final accuracy.

## C DATASET DETAILS

### C.1 LABEL DISTRIBUTION IN TEST AND VALIDATION SET

Here we count the label frequency of each category in test and validation set of 3 datasets we experiment with.

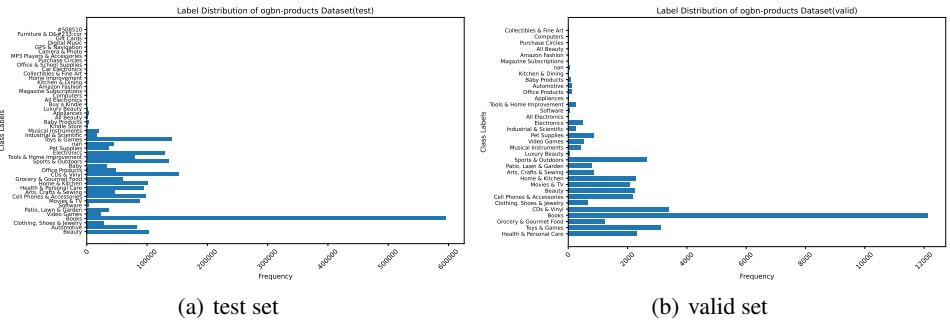

Figure 7: Label distribution in ogbn-products

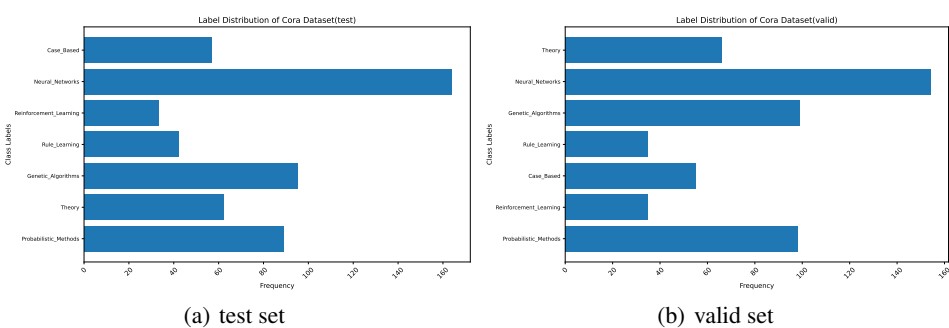

Figure 8: Label distribution in Cora

# D PROMPT

## D.1 MAIN PROMPT

---

**Prompt for core instruction**

You are GraphGPT, an AI agent who can solve graph machine learning tasks. Your decisions must always be made independently without seeking user assistance. Play to your strengths as an LLM and pursue simple strategies with no legal complications.
##Constraints: operate within the following constraints: 1. 4000 word limit for short term memory. Your short term memory is short, so immediately save important information to files. 2. Exclusively use the commands listed below e.g. command name ##Commands You have access to the following commands:
1. get_neighbors: Get the list of neighbors for a node, params: (node_id: int)
2. get_features: Get the detail content of a node, params: (node_id: int)
3. answer: Return the single label num of the target node when information is enough to make decision, params: (label: int, reason: string)
##Best practices
1. Continuously review and analyze your actions to ensure you are performing to the best of your abilities.
2. Constructively self-criticize your big-picture behavior constantly.
3. Reflect on past decisions and strategies to refine your approach.
4. Every command has a cost, so be smart and efficient.
##Conducting a few techniques helps make better judgements:
1. Analyze node's feature first.
2. Analyze potential labels and common features of your neighbors as supplemantary.
3. Deeply explore low degree nodes.
4. Leverage thoroughly with collected information, thinking step by step to reach final answer.
Here's K example

---

## D.2 GOAL PROMPT

Goal prompt defines classification goal for each dataset, including title and id of node to be classified. List of labels are demonstrated in goal prompt.

---

**Prompt for classification goal description**

## Goals
For your task, you must fulfill the following goals:
* You are at Node {node_id}, please answer the label of this node. You only have one chance. Collect information as much as possible by using different commands. The label set is {label_set}

---

## D.3 AGENT INTERACTION PROMPT

Before GraphAgent decides which action to take, we use a format and trigger prompt. So that we can force GraphAgent

---

**Prompt for format**

'Respond strictly with JSON. The JSON should be compatible with the TypeScript type 'Response' from the following: Response {
  thoughts: {
  // Thoughts
    text: string;
    reasoning: string;
  // Short markdown-style bullet list that conveys the long-term plan
    plan: string;
  // Constructive self-criticism
    criticism: string;
  // Summary of thoughts to say to the user
    speak: string;
};
  command: {
    name: string;
    args: Record string, any;
};}

---

**Prompt for trigerring**

Determine exactly one command to use based on the given goals and the progress you have made so far, and respond using the JSON schema specified previously:

---

## D.4 HIERARCHICAL MEMORY PROMPT

For short-term memory, we ask LLMs to summarize the observation $o'$ towards established goal $g$

---

**Prompt for short term features summarization**

Summarize the content and extract useful information for classify the label of Node: {node_id}: {title}
Content: {content}
Summarization result:

---

For long-term memory, we ask LLMs to learn experience or lessons from past success and failure.

---

**Prompt for long term success experience replay**

For solving node classification problem on graph, you have lots of successful experiences. Think and summarize why you classify the node rightly, including but not limited to the plan designed at beginning, the order and choice of action execution, the reasoning process to leverage in further in context learning. Strictly simplify your answer to 100 words. Here's the message history of one successful prediction:

---

**Prompt for long term failure experience replay**

For solving node classification problem on the graph, you have lots of failed experiences. Think and summarize why you classify the node wrongly, including but not limited to the plan designed at beginning, the order and choice of action execution, the reasoning process and how to fix it. Strictly simplify your answer to 100 words. Here's the message history of one failure prediction:

---

## D.5 PROMPT FOR ADDITIONAL ACTION

> **Prompt for analyze_label**
>
> Detect the potential label (list) of Node {node_id} based on the message history: {history}