# OpenReview forum: "GraphAgent: Exploiting Large Language Models for Interpretable Learning on Text-attributed Graphs"
_ICLR.cc/2024/Conference — Submitted to ICLR 2024_

### Official Review · Reviewer_985C · 2023-10-30

**Soundness:** 2 fair
**Presentation:** 2 fair
**Contribution:** 2 fair
**Rating:** 3
**Confidence:** 4

**Summary:**

This paper introduces GraphAgent, a framework for node classification on text-attributed graphs. The proposed method leverages the power of LLMs to parameterize the agent. Extensive experiments conducted on various datasets underscore the effectiveness and interpretability of GraphAgent.

**Strengths:**

1.	Exploring the power of LLMs for graph learning tasks is interesting.

2.	The illustrated case study and examples, as depicted in Figure 2 and Figure 5, are intriguing and contribute to improving interpretability.

**Weaknesses:**

1.	The authors claim to utilize LLMs to parameterize the graph agent, but they do not explicitly demonstrate the process of how to parametrize. It appears that LLM is inherently a pre-parametrized agent.

2.	The framework seems to be an implementation of the chain-of-thought paradigm applied to LLMs in the context of graphs. Therefore, the novelty is limited. Also another existing work explored the graph agent [1], it is helpful to make comparisons and discussions about the differences and connections with it.

3.	Can the proposed paradigm easily applied to other graph-related work, e.g., link prediction and graph classification. Can you provide some experimental results on these tasks.

4.	Unclear notations: $s_v$ in section 3.1 is not defined and explained; the multiple uses of $i$ in $y_i$ and $c_i$ may lead to confusion.

5.	Comparison with graph models that leveraged LLMs, e.g., [2-3], is necessary.

[1] Qinyong Wang, Zhenxiang Gao, and Rong Xu. Graph Agent: Explicit Reasoning Agent for Graphs. arXiv preprint arXiv:2310.16421 (2023).

[2] Ruosong Ye, Caiqi Zhang, Runhui Wang, Shuyuan Xu, and Yongfeng Zhang. Natural language is all a graph needs. arXiv preprint arXiv:2308.07134, 2023.

[3] Jianan Zhao, Le Zhuo, Yikang Shen, Meng Qu, Kai Liu, Michael Bronstein, Zhaocheng Zhu, and Jian Tang. Graphtext: Graph reasoning in text space. arXiv preprint arXiv:2310.01089, 2023.

**Questions:**

see weakness

---

### Official Review · Reviewer_Xvd3 · 2023-11-01

**Soundness:** 2 fair
**Presentation:** 2 fair
**Contribution:** 2 fair
**Rating:** 5
**Confidence:** 4

**Summary:**

This paper proposes a method GraphAgent which uses an agent to solve node classification tasks on TAGs. The agent is allowed to use actions to get the neighbors and features of the target node. The agent predicts the node category based on the information received. The agent follows an in-context learning scheme, where the examples in the prompt are randomly sampled. The experiments show that the method can improve GPT's ability to solve node classification tasks.

**Strengths:**

1. Using agent to solve node classification tasks on TAGs by asking for neighborhood information is a novel idea.
2. The designs about actions make sense, using random sampling to generate examples to enable in-context learning is smart.
3. The experiment cases show that the LLM can use the designed actions to get useful information to give the prediction.

**Weaknesses:**

1. The case shown in Figure 2 is not clear. Which part is the output of the LLM and which part the given by the environment? Besides, Section 4.1 designs three types of actions, but their formats are not consistent. I can find "get features" and "get neighbors" as actions but the answer is given by natural language. Please clarify your method.
2. According to [1], GPT3.5 can achieve 67% accuracy on Cora, 50% on arxiv, 70% on Product. But table 2 shows it can only achieve very low performance (gpt-3.5-turbo + ICL). I guess it is because of the prompt used. I suggest to use a better prompt for the baseline.
3. The performance of the method cannot outperform GNN based methods.

[1] Chen, Z., Mao, H., Li, H., Jin, W., Wen, H., Wei, X., ... & Tang, J. (2023). Exploring the potential of large language models (llms) in learning on graphs. arXiv preprint arXiv:2307.03393.

**Questions:**

1. The description of section 4.3 is not very clear. What is the difference between the short-term memory and the long-term memory? I.e., when you get memory_short and memory_long, how to put them in the memory/context? Could you show them in a case?

---

### Official Review · Reviewer_39D5 · 2023-11-01

**Soundness:** 4 excellent
**Presentation:** 3 good
**Contribution:** 3 good
**Rating:** 5
**Confidence:** 3

**Summary:**

This paper proposes GraphAgent, which uses large language models to frame learning on text-attributed graphs as an agent planning problem. This provides interpretability and enables explicit actions tailored for graph structure and node texts. Experiments show GraphAgent effectively captures relationships to make precise, interpretable predictions, demonstrating the promise of combining large language models with graph learning.

**Strengths:**

* The idea of using a chain-of-thought approach to allow large language models to automatically reason through defined graph-related actions is novel. It can bring new ideas to the community on how to consider the application of LLM on graphs.

* The experiment results, compared to other baselines, prove that GraphAgent can compete with supervised learning without the need for an external GNN. This result can inspire more work to notice the new opportunities that large models bring to graph reasoning tasks.

**Weaknesses:**

* The description of the agent prompt can only be found in Appedix, making it difficult to understand the author's approach without referring to the supplementary materials.
* There is a lack of efficiency comparison. Intuitively, actions allow LLMs to see more node information, but processing this information also incurs additional costs. However, the paper's experiments do not discuss this issue.
* There is a lack of discussion on stability. I am concerned about whether small perturbations can make the agent unstable, which would greatly affect the reliability and reproducibility of the experimental work.

**Questions:**

Can you show the cost of accessing each dataset through GraphAgent for processing with ChatGPT?

**Details Of Ethics Concerns:**

None.

---

### Official Review · Reviewer_sBxi · 2023-11-04

**Soundness:** 2 fair
**Presentation:** 2 fair
**Contribution:** 1 poor
**Rating:** 3
**Confidence:** 5

**Summary:**

This paper introduces GraphAgent, an in-context learning framework that employs an LLM as the agent to gather information and predict node labels in text-attributed graphs. Specifically, this paper formalizes the node classification task as an agent-planning task, with pre-defined actions and a memory mechanism. Few-shot experiments are conducted on Cora, ogbn-arxiv, and ogbn-products to evaluate the performance. Yet, it exhibits inferior efficacy when juxtaposed with MLP- and GNN-based conventional methods.

**Strengths:**

Given the emergence of LLMs, it’s worth exploring their potential application in the graph field to enhance graph comprehension, especially on topological structures. The authors apply LLM for text-attributed graph understanding.

**Weaknesses:**

1. The experimental results are insufficient to substantiate claimed contributions. Specifically,  the test is conducted on only 50 samples, which significantly undermines the credibility of the experimental outcomes. Concurrently, the interpretability mentioned by the authors lacks further elucidation and quantitative evaluation.

2. The performance gap between GraphAgent and conventional methods is still large. In fact, 9.7% in ogbn-arxiv, and a 6.5% margin in ogbn-products indicate a large gap instead of comparable performance. Not to mention the computational efficiency.

3. Possible test data leakage. The training corpus of GPT includes papers from arXiv, implying that the GPT model may already be aware of the test node labels, that is, the paper's field. This can be tested by directly asking GPT about a paper’s field, given its text-attributes (title and abstraction). The authors are encouraged to conduct this experiment to discuss the possible test data leakage problem.

4. Notations in Section 3.1 are really confusing. For example, what is f_v? y_L denotes “a few labeled node” while y_U are “labels of unlabeled nodes”. “K is the number of labeled nodes of each category” Does each category necessarily have the same number of labeled nodes K?

5. The writing needs to be polished and the authors need to be more careful with their claims. For instance, “For this goal, we introduce GraphAgent as our solution to leverage Large Language Models (LLMs) for graph machine learning tasks, with a specific focus on node classification due to its wide application.” Based on my understanding, GraphAgent might be a solution for the node-classification task in text-attributed graphs, instead of a solution for graph machine-learning tasks with a focus on node classification.

6. It might be better to place the tables and figures on the top of each page.

**Questions:**

See weakness part for my major concerns. I am curious about the benefits of GraphAgent in comparison to conventional methods for text-attributed graph comprehension. Performance, computational efficiency or interpretability?

---

### Meta-Review · Area_Chair_9RQ3 · 2023-12-09

**Metareview:**

This paper proposes GraphAgent, a method to use LLMs for node classification tasks. The reviewers find that using LLMs for graph learning tasks is interesting; however, there are multiple weakness that need to be addressed before publication, including more extensive experiments study and comparisons with baselines, and questions on the effectiveness as the performance is not comparable to GNN methods. The authors did not submit a rebuttal in response to the reviewers concern. Therefore, I recommend rejection for this paper.

**Justification For Why Not Higher Score:**

The quality of this paper does not meet ICLR standard.

**Justification For Why Not Lower Score:**

N/A

---

### Decision · Program_Chairs · 2024-01-16

Reject